# The Host Immune Response to *Scedosporium*/*Lomentospora*

**DOI:** 10.3390/jof7020075

**Published:** 2021-01-22

**Authors:** Idoia Buldain, Leire Martin-Souto, Aitziber Antoran, Maialen Areitio, Leire Aparicio-Fernandez, Aitor Rementeria, Fernando L. Hernando, Andoni Ramirez-Garcia

**Affiliations:** Fungal and Bacterial Biomics Research Group, Department of Immunology, Microbiology and Parasitology, Faculty of Science and Technology, University of the Basque Country (UPV/EHU), Barrio Sarriena s/n, 48940 Leioa, Spain; idoia.buldain@ehu.eus (I.B.); leire.martin@ehu.eus (L.M.-S.); aitziber.antoran@ehu.eus (A.A.); maialen.areitio@ehu.eus (M.A.); leire.aparicio@ehu.eus (L.A.-F.); fl.hernando@ehu.eus (F.L.H.)

**Keywords:** *Scedosporium*, *Lomentospora*, immune response, cytokine, macrophage, lymphocyte, monocyte, antigen, epithelium

## Abstract

Infections caused by the opportunistic pathogens *Scedosporium*/*Lomentospora* are on the rise. This causes problems in the clinic due to the difficulty in diagnosing and treating them. This review collates information published on immune response against these fungi, since an understanding of the mechanisms involved is of great interest in developing more effective strategies against them. *Scedosporium*/*Lomentospora* cell wall components, including peptidorhamnomannans (PRMs), α-glucans and glucosylceramides, are important immune response activators following their recognition by TLR2, TLR4 and Dectin-1 and through receptors that are yet unknown. After recognition, cytokine synthesis and antifungal activity of different phagocytes and epithelial cells is species-specific, highlighting the poor response by microglial cells against *L. prolificans*. Moreover, a great number of *Scedosporium*/*Lomentospora* antigens have been identified, most notably catalase, PRM and Hsp70 for their potential medical applicability. Against host immune response, these fungi contain evasion mechanisms, inducing host non-protective response, masking fungal molecular patterns, destructing host defense proteins and decreasing oxidative killing. In conclusion, although many advances have been made, many aspects remain to be elucidated and more research is necessary to shed light on the immune response to *Scedosporium*/*Lomentospora*.

## 1. Introduction

Among microbial pathogens, fungi have historically been treated as a minor threat to human health compared to other infectious agents such as viruses or bacteria, because they affected a lower percentage of patients and antifungal treatments were effective for most of them. In recent decades, however, the outlook has been changing radically, as fungal infections increase in frequency as a result of a substantial increase in immunosuppressive infections, such as those generated by the human immunodeficiency virus (HIV), as well as in the use of more potent immunosuppressive drugs and in invasive medical interventions. Moreover, resistance to drugs has become more and more common, which may be due to several reasons, including the use of environmental fungicides as reported in *Aspergillus* resistance to azole [1].

Even amongst pathogenic fungi, there has been a discriminatory behavior in favor of genera such as *Candida*, *Cryptococcus* or *Aspergillus*, because of the higher incidence of infections they cause compared to other fungi [2,3,4,5]. However, rare mycoses, especially those caused by filamentous fungi, are on the rise [5], probably as a result of the use of antifungal prophylaxis in all patients at risk of opportunistic fungal infection. This routine has proven effectivity in preventing the most frequent molds [6] but may be contributing to the emergence of less common mold infections such as those caused by the *Lomentospora* and *Scedosporium* species, which are highly resistant to current antifungal drugs [7,8].

The species of the genera *Scedosporium* has been reclassified in recent years. The genus currently includes 10 species [9,10,11] formerly classified within the *Pseudallescheria*/*Scedosporium* complex; the dual name was replaced by the genus *Scedosporium* following the “Amsterdam Declaration on Fungal Nomenclature”, which recommends a single-name system for all fungi [12]. Some years later, the species *Scedosporium prolificans* was separated and redefined as *Lomentospora prolificans* [13].

The *Scedosporium*/*Lomentospora* species are saprophytes usually found in soil, especially from human-disturbed ecosystems [14], but can also behave as opportunistic pathogens. They are considered as emerging fungal pathogens due to the increasing frequency of infections associated with high mortality rates. These infections appear mainly as a consequence of direct inoculation, respiration, and aspiration of polluted water during near-drowning events, with a remarkable prevalence after tsunami catastrophes [9,11,15,16]. In any case, once the infection has disseminated, systemic infections lead to a worse scenario. They also show neurotropism, meaning that central nervous system (CNS) infections are a probable consequence [9,17].

Specifically, they can cause a wide variety of diseases from localized infections in immunocompetent individuals to disseminated infections in those that are immunocompromised [14]. Within the immunocompromised group, transplant recipients and patients with primary hematological malignancies are among those at highest risk of suffering from these mycoses [9,11,15,18,19]. In the case of immunocompetent individuals, those who suffer from cystic fibrosis (CF) are susceptible to chronic airway colonization. In these patients, *Scedosporium*/*Lomentospora* species show a prevalence ranging between 0–25%, ranking second among filamentous fungi, after *Aspergillus fumigatus* [9,11,15,20,21,22,23,24]. This colonization is considered a risk factor for the development of an infection [24] and contributes to chronic inflammation, which can lead to the progressive deterioration of the lung function, as demonstrated for *A. fumigatus* [24,25,26].

Diagnosis is still based on culture-dependent methods and serological tests are only performed on a “home-made” basis, [27,28], but are not commercially available. This makes it impossible to differentiate between an airway colonization and a respiratory infection. Treatment is difficult, as these fungi show very limited susceptibility to current antifungals. European guidelines currently recommend voriconazole as a first-line treatment, together with surgical debridement when possible. In some cases, combinations of antifungals are needed, of which the most studied are voriconazole and terbinafine [9,29,30,31,32]. However, combination therapy is only supported by some reports and is therefore only moderately recommended.

Considering the rise in cases of *Scedosporium*/*Lomentospora* and the problematic nature of their diagnosis and treatment, scientific advances are needed. In this regard, an understanding of all the immune mechanisms involved in a successful response against these pathogens is of paramount importance. This review therefore compiles the information published over recent years to explain this clearly and to make comparisons with other filamentous fungi.

## 2. Pathogen Recognition

As in most microorganisms, activation of an innate immune response to fungal infections is induced by the recognition of molecular components shared by many fungal species, found mainly on the cell wall, known as Pathogen Associated Molecular Patterns (PAMPs). To achieve this, the cells in the immune system use Pattern Recognition Receptors (PRRs) which recognize microbial molecules and activate signaling pathways that generate induction of the synthesis of pro-inflammatory cytokines, phagocytosis and adaptive immunity [33,34], as described below. The PRRs most implicated in the recognition of fungal pathogens are usually Toll-like receptors (TLRs), C-type Lectin Receptors (CLRs), Dectin-1 and 2, and Mannose Receptor/CD206 (MR) [33]. Specifically, the recognition of *Scedosporium*/*Lomentospora* has not yet been described in depth, but some of these PRRs have been indicated as being of paramount importance (Figure 1 and Figure 2, with references in Appendix A, respectively).

### 2.1. Toll-Like Receptors: TLR2 and TLR4

As already mentioned, TLRs are a family of glycoproteins that act as molecular sensors, mediating early recognition of a large array of conserved microbial molecules and crucial for initiating the innate response against many microorganisms. The structure of the typical TLR contains three domains: a leucine-rich repeat (LRR) motif responsible for pathogen recognition, a transmembrane domain, and a cytoplasmic Toll/IL-1 receptor (TIR) domain to initiate signal transduction. Since the first report of a Toll protein in *Drosophila melanogaster*, 10 TLRs have been identified in humans (TLR1–TLR10) and 13 in mice (TLR1–TLR13) [35]. Among these, TLR2 and TLR4—the best characterized TLRs—have been widely described as crucial for recognizing fungal PAMPs—specifically, polysaccharides such as the glucuronoxylomannans (GXMs) of the *Cryptococcus* capsule, recognized by TLR4 and complexes TLR2/TLR1 and TLR2/TLR6 [36,37] and the mannans from *Candida albicans* and *Saccharomyces cerevisiae*, recognized by TLR4/CD14 [38]. Indeed, lack of TLR2 and TLR4 leads to an increased susceptibility to *A. fumigatus* and *C. albicans* in mice [39,40,41,42]. Similarly, an essential role of these receptors has also been demonstrated in response to *Scedosporium*/*Lomentospora*; while wild-type *Drosophila melanogaster* flies are resistant to infection with *S. apiospermum* and *L. prolificans*, TLR-deficient flies develop acute infections with high mortality rates [43].

More specifically, TLR2, together with CD14, has been described as the PRR responsible for recognizing *S. boydii* α-glucans and inducing a signaling pathway through the adapter protein of myeloid differentiation primary response gene 88 (MyD88), which stimulates pro-inflammatory cytokine secretion in peritoneal murine macrophages (TNF) (Figure 1) and bone-marrow derived dendritic cells (IL-12 and TNF) (Figure 2A) [44]. Surprisingly, although α-glucans are involved in phagocytosis, the specific receptor involved in this process has yet to be elucidated.

On the other hand, TLR4 binds to *S. boydii* rhamnomannans, which are strongly expressed on the cell surface of these fungi, resulting in the release of cytokines (TNF, IL-6, IL-10 and IP-10/CXCL10) by macrophages (Figure 1) [45]. Specifically, removal of rhamnopyranosyl prevents cytokine release, implying that structures with terminal rhamnose and/or mannose are involved in TLR4 recognition, while de-O-glycosylated peptidorhamnomannans induce the same cytokine release as that obtained by total rhamnomannans. The role of TLR4 in *S. boydii* recognition is so important that this receptor, together with CD14 and MyD88 (but not TLR2) is required to induce macrophage activation by conidia (Figure 2B). Surprisingly, hyphae recognition is independent of both TLR2 and TLR4, so the role of TLR in hyphae recognition is unknown [45].

In summary, both TLR2 and TLR4 appear to take part in *Scedosporium* conidia recognition, by binding to fungal cell-wall α-glucans and rhamnomannas, respectively. However, while TLR4 plays an essential role in macrophage activation by fungal conidia, this activation is independent of TLR2. It seems contradictory that while on the one hand, *S. boydii* α-glucans induce the release of pro-inflammatory cytokines in macrophages through TLR2, on the other hand, macrophage activation by *S. boydii* is TLR2-independent. This finding therefore proves that the role of TLR2 in *Scedosporium* spp. recognition and immune response activation has yet to be elucidated.

### 2.2. C-Like Receptors: Dectin-1 and Mannose Receptor

C-type lectin receptors (CLRs) comprise a large family of receptors that bind to carbohydrates in a calcium-dependent manner and one group of soluble and two groups of membrane-bound CLRs can be distinguished, based on their molecular structure. The most representative receptors against fungal infections are Dectin-1 and the mannose receptor (MR), among the Type I and II membrane-bound CLRs, respectively.

On the one hand, C-type lectin 1 (Dectin-1) binds to fungal β (1,3)-glucans, and its activation results in a protective antifungal immunity through crosstalk of two independent signaling pathways—one through spleen tyrosine kinase (SYK) and the other through RAF1—which are essential for the expression of Th1 and Th17 cell polarizing cytokines [46]. On the other hand, MR recognizes fungal wall mannans and is involved in both homeostatic processes and pathogen recognition. However, although MR has been shown to be essential for cytokine production, its function in host defense is not yet clearly understood, since MR-deficient animals do not display enhanced susceptibility to pathogens bearing MR ligands [47].

Concerning *Scedosporium*/*Lomentospora*, an increase has been demonstrated in the hyphal damage induced by polymorphonuclear cells (PMN) after recognition of *S. apiospermum* β-glucans, while this effect has not been observed for *L. prolificans* [48]. Indeed, the increase in fungal cell wall β-glucan exposure after caspofungin treatment was associated with a significant increase in PMN-mediated hyphal damage against *Aspergillus* spp., *Fusarium* spp. and *S. apiospermum*, but not *L. prolificans*. This effect was further augmented by the addition of anti-β-glucan antibody. Backing these results, it has been observed that pre-exposure of *A. fumigatus*, but not *L. prolificans*, to caspofungin increases the expression of Dectin-1 in PMNs [48].

However, in a study performed by Pellon et al. (2018) it was shown that MR, but above all Dectin-1, mediates *L. prolificans* conidia uptake by microglial cells (Figure 2C) [49]. Particularly, individual blocking of each receptor successfully inhibited phagocytosis of the fungus and, more interestingly, the double inhibition of the receptors did not lead to a synergistic effect, with similar values being obtained as when only Dectin-1 was inhibited. Furthermore, under no experimental conditions was phagocytosis completely inhibited, showing that other receptors may be involved in this process [49].

### 2.3. Fungal Ligands with Unknown Immune Cell Receptors

Important molecules of the *Scedosporium*/*Lomentospora* fungal cell wall, such as peptidorhamnomannan (PRM) and glucosylceramide, are involved in fungal recognition by mechanisms that are still unknown (Figure 1). This section describes the studies that demonstrate the involvement of these surface molecules in *Scedosporium*/*Lomentospora* recognition.

Peptidorhamnomannan is present on the surface of both conidia and mycelia of *S. boydii* and *S. apiospermum*, and on the conidial surface of *L. prolificans* and *S. aurantiacum* [50,51,52]. PRM from *S. boydii* is involved in conidia adhesion to and infection of epithelial cells, through binding to a non-identified 25 kDa polypeptide on the epithelial cell surface [53]. Furthermore, PRM from *S. apiospermum* is involved in fungal phagocytosis by macrophages, since the addition of anti-PRM mAbs reduced conidial phagocytosis by macrophages and superoxide production [52]. Regarding *L. prolificans*, PRM participates in fungal recognition and phagocytosis, induces macrophage death, and stimulates production of pro-inflammatory cytokines (TNF) and nitric oxide (NO) and, precisely, the O-glycosides of the molecule appear to be responsible for these effects [54]. Along the same lines, a recent study has shown that PRM from *S. aurantiacum* is involved in fungal phagocytosis and also increases in NO production by macrophages [50].

On the other hand, glucosylceramide (GlcCer) has been described as a potent immune response activator present on the *Scedosporium*/*Lomentospora* conidial and hyphal cell surface [55]. Specifically, purified GlcCer from *S. aurantiacum* and *L. prolificans* leads to peritoneal macrophage activation, inducing NO and superoxide production and increasing the killing of *L. prolificans* conidia [56,57]. In addition, in vivo assays have shown that purified GlcCer species from *L. prolificans* were able to induce a high secretion of pro-inflammatory cytokines by splenocytes, and to promote the recruitment of PMNs, eosinophils, small peritoneal macrophages and mononuclear cells to the peritoneal cavity [55,57]. However, the GlcCer receptor/s have not yet been elucidated.

Concerning α-glucan, in addition to stimulating innate response cells through TLR2, CD14 and MyD88 as explained above [44], it is also involved in *S. boydii* uptake by macrophages. Indeed, addition of soluble α-glucan generates a dose-dependent inhibition of conidia phagocytosis, and the enzymatic removal of this molecule from the conidia surface decreases the phagocytic index [44]. Therefore, α-glucan has an essential role in conidia recognition and internalization by macrophages, but the receptor involved in this process has not as yet been identified. This role was not observed for fungal β-glucans although in other fungi, such as *C. albicans* and *Saccharomyces*, phagocytosis by macrophages is critically dependent on the recognition of surface-associated β-glucans by Dectin-1 [44].

## 3. Activation of Innate Immune Cells

Recognition of fungal molecules leads to an activation of immune cells to respond against the microbial presence. It is obvious that the pathways activated are different depending on the fungal molecule recognized and the receptor used. However, the type of immune cell is also a determining factor. The following paragraphs summarize current knowledge on the response given by monocyte-derived macrophages, peritoneal macrophages, neutrophils, microglial cells, and epithelial cells against *Scedosporium*/*Lomentospora* (Figure 3, with references in Appendix A).

### 3.1. Mononuclear Cells and Polymorphonuclear Phagocytes

Phagocytes are one of the most important cells in the immune system for countering fungal infections because of their ability to detect and phagocytose them, but also for promoting pro-inflammatory signals to attract and activate other immune cells [58]. Phagocytes are classically classified into two main groups: mononuclear cells (MNCs), which are monocytes and macrophages; and PMN phagocytes, also known as neutrophils.

In addition to the capacities described above, macrophages are also important because they can act as antigen presenting cells (APC) and thus connect innate and adaptive immune responses. Most of the macrophages are obtained from circulating monocyte differentiation when they infiltrate the tissues from blood vessels. However, there are many populations of “resident” macrophages in specific organs, which have a fetal origin and are maintained by their homeostatic multiplication. These macrophages include, inter alia, cerebral microglial cells, liver Kupffer cells, lung alveolar cells, and splenic and renal macrophages [59].

Different types of macrophages have shown an ability to phagocytose *Scedosporium*/*Lomentospora* species. In the case of peritoneal macrophages, they have been shown to be able to phagocyte *Scedosporium*/*Lomentospora* species in a species-specific manner [50]. Specifically, *S. aurantiacum* and *L. prolificans* conidia are more efficiently phagocyted than *S. minutisporum*, although they are less likely to be killed. This suggests that *S. aurantiacum* and *L. prolificans* are provided with mechanisms that enable them to survive inside macrophages [50].

Furthermore, monocyte-derived macrophages are able to phagocyte *L. prolificans* conidia, in a manner comparable to *Aspergillus*, despite the much larger size of its conidia. However, they inhibit the germination of *L. prolificans* conidia less efficiently than that of *A. fumigatus* [60]. Confirming these results, even if phagocyted, *S. apiospermum* and *L. prolificans* conidia can germinate within macrophages, forming germ tube-like projections that can lyse the membrane at this point to reach the extracellular medium [49,61].

If the *Scedosporium*/*Lomentospora* has germinated, phagocytosis becomes difficult and, therefore, not only circulating monocytes attracted to the infection site, but also neutrophils, become essential [9]. Indeed, although primary macrophages can also damage hyphae, the bulk of the innate defense against hyphae appears to fall upon the exo-cytose capacity of the neutrophils. These PMNs damage hyphae mainly by degranulation, release of large amounts of reactive oxygen species (ROS), and the formation of neutrophil extracellular traps (NETs), which trap fungal cells in a matrix mainly composed of DNA and proteins with antimicrobial activity [9,62]. Moreover, when *L. prolificans* hyphae are opsonized, superoxide production by PMN and circulating MNC is significantly stimulated, increasing their antifungal effect [60]. In this way, MNC and PMN tend to damage *L. prolificans* hyphae to an equal and greater degree, respectively, than *A. fumigatus* hyphae [60]. This susceptibility of *L. prolificans* to these cells may be related to the very low pathogenicity of the mold in healthy individuals and its high incidence in neutropenic patients [18,63].

### 3.2. Microglia

As the resident macrophage cells in the central nervous system (CNS), microglial cells act as the first and main form of active immune defense against microorganisms there. However, their role in countering different species of *Scedosporium*/*Lomentospora* appears to differ greatly. Indeed, microglial cells bring impaired phagocytic capacity in *L. prolificans*, as well as lower levels of pro-inflammatory cytokine release (TNF and IL-6) and ROS production, as compared to other phagocytes [49]. Specifically, experiments using the related species *S. boydii* and *S. aurantiacum* and the high-prevalence yeast *C. albicans* showed that *L. prolificans* and *C. albicans* were respectively the least and most efficiently phagocytized [49].

Moreover, like other phagocytes studied, extreme acidic environments may be found inside microglial phagolysosomes, but *L. prolificans* cells are able to survive pH stress, maintaining high levels of viability in both basic and acidic conditions [49]. This characteristic may allow fungal cells to persist inside phagolysosomes. Along the same lines, it was observed that, as the hours went by, engulfed conidia were germinated inside microglial cells and appear to be able to pierce cell membranes, as indicated above for other types of phagocyte. This activity may contribute to the observed microglial cell death induced by fungal cells [49]. All these data demonstrating a poor microglial response against *L. prolificans* might at least partly explain the tendency of this fungus to infect the CNS, a phenomenon known as neurotropism [9,17].

### 3.3. Non-Professional Phagocytes: Epithelial Cells

Epithelial barriers are one of the first and most important mechanisms for preventing entrance of microorganisms into the host. The epithelia constitute a physical and chemical barrier, since many antimicrobial substances are secreted, but also a biological barrier, since they are in direct contact with microbiota, which helps host control pathogens. Epithelial cells are the most abundant cells in these barriers, which constitute the skeleton of these tissues. These cells are able to bind and respond against microorganisms using different mechanisms, including phagocytosis in several types of epithelial cells [64].

One of the most important routes used by *Scedosporium*/*Lomentospora* species to colonize or infect the host is through the airways [18,65]. Taking this into consideration, and also the fact that there is increasing evidence of a key role for the airway epithelium in the response to respiratory pathogens, particularly at early stages of fungal infection [66], studies are presented below explaining the response of two different epithelial cell lines against *Scedosporium*/*Lomentospora*.

#### 3.3.1. Larynx Carcinoma Cells

Despite the fact that HEp2 human larynx carcinoma cells are considered to be non-professional phagocytic cells, conidia of *S. boydii* are attached to and ingested by them, suggesting active participation by this fungus in the interaction process. Indeed, conidia produce an early germ-tube like projection in the presence of epithelial cells, which is capable of penetrating the membranes of these cells’ membrane after only 4 h of co-incubation [53].

To be more precise, the binding of *S. boydii* conidia to epithelial cells is mainly mediated by the PRM present on the surface of the fungus, since pre-treatment of HEp2 cells with soluble PRM resulted in 50% and 60% lower adhesion and endocytic index, respectively, than untreated epithelial cells [53]. Along the same lines, cell pre-incubation with anti-PRM antibody inhibits conidia binding and endocytosis in a dose-dependent manner. Moreover, it has been identified that the PRM of *S. boydii* binds to a 25 kDa polypeptide on the HEp2 cell surface and it seems clear that the carbohydrate fraction of the PRM, such as mannose, plays an important role, especially in binding events [53].

#### 3.3.2. Human Lung Epithelial Cells

Conidia of *Scedosporium*/*Lomentospora* are able to adhere, differentiate into hyphae and form biofilms on human lung epithelial A549 cells, which implies, in a similar way as mentioned above for human larynx epithelial cells, an active participation of the fungus in this interaction [61,67,68]. Specifically, *S. aurantiacum* germinating conidia were able to invade epithelial cells through the intercellular space without intracellular uptake of fungal conidia. This contrasts with studies with *A. fumigatus*, in which a rapid internalization of fungal conidia was observed after a first binding step. After 24 h of co-incubation, A549 cells were completely detached from the culture plate [67]. Moreover, another study observed that the interaction of A549 cells with *S. apiospermum*, *S. minutisporum* and *S. aurantiacum* led to biofilm structures covering epithelial cells, while *L. prolificans* cells completely destroyed the monolayer of the epithelial cells [61,68].

On the other hand, transcriptomic analysis of A549 cells infected with live *S. aurantiacum* conidia showed an up-regulation of genes implicated mainly in cell repair and inflammatory processes, whereas down-regulated genes were mainly involved in cell cycle progression, as happens with *A. fumigatus* [67]. Furthermore, network analysis revealed an activation of the innate immune system through the NF-kB pathway which might be involved in the up-regulation of IL-11 and CXCL8/IL-8 pro-inflammatory cytokines in the presence of the fungi [67].

### 3.4. Cytokine Release

Cytokines are small (15–20 kDa) and short-lived proteins that coordinate the development and activity of the immune system by autocrine, paracrine and endocrine signaling [69]. Many of them are secreted in response to fungal recognition by PRRs and infections, but the release of only some of these small signal molecules against *Scedosporium*/*Lomentospora* species has so far been studied.

#### 3.4.1. Cytokines Released against *Scedosporium*/*Lomentospora*

The cytokines synthesized may differ depending on the fungal species and the type of cell stimulated. In the case of *Scedosporium* species, *S. boydii* induce a strong activation of murine peritoneal macrophages, which lead to the secretion of substantial amounts of IL-10, IL-12, IP-10/CXCL10, TNF and low IL-6 against conidia, whereas hyphae induce higher TNF and IL-6 release, but not IL-10 [45]. The differential induction of IL-10 according to the morphology has also been described for *A. fumigatus* and *C. albicans* [70,71] and may be related to pathogenesis. Indeed, taking into account that IL-10 is an anti-inflammatory cytokine involved in inhibiting macrophage activation, induction of IL-10 by conidia may generate an initial microenvironment which facilitates their germination in tissues [45].

In the case of human monocytes, they produce significantly more IL-6 and TNF when challenged in vitro with *L. prolificans* conidia than with *Aspergillus* spp., including *A. fumigatus*, and *Rhizopus arrhizus* [70]. The components of the fungal cell wall of *L. prolificans* are largely unknown, but it may contain molecules with higher stimulatory capacity to induce TNF and IL-6 release. Likewise, Pellon et al. (2018) demonstrated IL-6 and TNF release in response to *L. prolificans* by peritoneal macrophage-like cells and microglial cells [49], although the macrophages produced them faster and at a higher concentration, which supports the previously mentioned hypothesis of an impaired microglial response against this fungus [49]. These studies highlight the relevance of the pro-inflammatory cytokines, such as IL-6 and TNF, in response to *Scedosporium*/*Lomentospora* species. This is not surprising as these cytokines have an especially important role in protection against infections, stimulating PMN migration, inducing respiratory burst by PMNs and PMN-mediated antibody-dependent cellular cytotoxicity [72,73]. Indeed, TNF is considered necessary for the development of effective innate and adaptive immunity to a wide variety of fungal infections by stimulating neutrophils and macrophages against other fungi such as *C. albicans*, *A. fumigatus* and *C. neoformans*, and also by inducing the secretion of other cytokines, including IFN-γ IL-1, IL-6 and IL-12 [74,75,76,77,78,79].

#### 3.4.2. Cytokines as Therapeutic Tools

Since cytokines are secreted by immune cells in response to microbial presence, several studies have analyzed their antifungal effect and therapeutic utility against *Scedosporium*/*Lomentospora* alone or, above all, in combination with other drugs.

In this regard, the combination of granulocyte colony stimulating factor (G-CSF) with antifungals has been demonstrated to be efficacious against *L. prolificans* infections [80,81]. G-CSF is a hematopoietic growth factor, which stimulates the proliferation and differentiation of myeloid progenitor cells to neutrophils and enhances their phagocytic activity against pathogenic fungi [79]. Specifically, clinical resolution of *L. prolificans* fungemia was described as being associated with the reversal of neutropenia following administration of G-CSF in combination with amphotericin B [80]. In addition, although G-CSF also appears to promote the ‘non-protective’ Th2 response [79], its administration improves survival in neutropenic animal models of invasive fungal infections (IFI) [82,83,84]. Indeed, focusing on disseminated infection by *L. prolificans*, G-CSF treatment combined with liposomal amphotericin B (LAMB) improved survival of immunosuppressed mice compared to LAMB alone, but was not statistically significant [81].

Other cytokines studied for therapy are the granulocyte-macrophage colony-stimulating factor (GM-CSF) and interferon gamma (IFN-γ). On the one hand, GM-CSF accelerates myeloid hematopoiesis to produce more neutrophils, monocytes and eosinophils [79], and augments their activity against fungal species [85,86,87,88,89], but also enhances TLR2 and Dectin-1 expression [90,91]. On the other hand, IFN-γ is a key cytokine both in the innate and adaptive immune response to IFI, mainly because it stimulates migration, adherence and antifungal activity of neutrophils and/or macrophages [76,79,85,88,92]. These two cytokines, particularly in combination, showed an ability to augment the antifungal activity of human PMN against *S. apiospermum* and *L. prolificans* by increasing superoxide production [93,94].

Finally, IL-15, which has been described as being involved in enhancing the antifungal activity of PMN, monocytes and even NK cells against other fungi [95,96,97,98], also significantly increases human PMN-induced *L. prolificans* hyphal damage, but not *S. apiospermum* [99]. In this case, the antifungal effect is not concomitant with an increase in superoxide production, suggesting that other mechanisms, such as antifungal peptides, are upregulated. Indeed, IL-15 increased IL-8 release, but not TNF, from PMNs, which might promote hyphal damage promoting recruitment and activation of PMNs, and antifungal peptide release.

## 4. Activation of Adaptative Immune Cells

### 4.1. T-Cell Response

Antigen-presenting cells—mainly dendritic cells—digest the potential antigens and present them to T cells, promoting differentiation of T-helper (Th), cytotoxic (Tc) and regulatory (Treg) cells depending on the stimulus and PRR involved. Specifically, to overcome fungal infections Th1, Th2 and Th17 cells are of utmost importance [9,100]. For example, In the case of the widely studied fungal pathogen *C. albicans*, Th1 and Th17 responses have been shown to be protective, whereas Th2 type response is related to a worse infection outcome [101]. However, in the case of *Scedosporium*/*Lomentospora* infections, there are almost no studies regarding T cells, and the data presented are not enough to establish the role of T cells in these infections. In a study conducted among HIV patients, Tammer et al. (2011) found that patients with proven invasive scedosporiosis presented lower CD4+ cell levels than patients that were only colonized [102]. However, the T cells are not the only CD4+ cells, and thus a direct correlation between invasive scedosporiosis and low T cell counts cannot be established. In addition, in mice infections conducted by Xisto et al. (2016), they found that Th17 levels did not significantly change when mice were infected with *L. prolificans*, although T memory cell populations did significantly increase, probably as a result of the ongoing infection [103]. Finally, another study showed that *L. prolificans* lysates were able to expand specific T cells for this fungus in vitro, mainly CD4+ cells (81% CD4+ vs. 9.5% CD8+) [104]. Taken together, these three studies might indicate that a T-cell response is activated against *Scedosporium*/*Lomentospora* infections, primarily a Th response other than Th17, but many more studies need to be conducted to elucidate fully the interaction of T cells with this species complex.

### 4.2. B Cell Response

An essential role of Th cells, in this case specifically of Thf cells (a population that tends to be underrated or even omitted from many studies) is the activation of antibody-producing B cells. Until 1990 antibody-mediated immunity was considered irrelevant in the host response to fungi, mainly due to the methods used thus far, such as passive transfer of immune serum, which could not be demonstrated to be effective. However, the technology of monoclonal antibody (mAbs) production has provided evidence for the role of specific antibodies to the benefit or detriment of the host [105]. The functions for combatting fungal pathogens include classical mechanisms such as phagocytosis and complement activation, but recent studies suggest additional mechanisms for antibodies, such as modulation of the inflammatory response, and inhibition of replication, germination, polysaccharide release, morphologic switch and biofilm formation, among others [105]. Likewise, protective antibodies against a wide variety of fungal pathogens, including *C. neoformans*, *C. albicans*, *Histoplasma capsulatum*, *A. fumigatus*, *Paracoccidioides brasiliensis* and *Sporothrix schenckii*, have also been described [106].

Different studies have demonstrated a strong and complex humoral response in immunocompetent and non-infected individuals to *L. prolificans*, mediated by both salivary IgA and serum IgG [107,108,109]. Specifically, salivary IgA recognizes almost exclusively the conidia of the fungus, while serum IgG recognizes both conidia and hyphae. This is consistent with the scenario of a fungal invasion of the respiratory tract, in which the host inhales the morphology used by the fungus for dispersal, the conidia, instead of hyphae [108]. Furthermore, taking into account the fact that some of the antigens identified in these studies are broadly conserved among fungal pathogens, this high reactivity observed for a healthy population may be related to a cross-reactivity process among these antigens and those of other common fungi, such as *A. fumigatus* [108,109].

These studies using sera and saliva from immunocompetent individuals are interesting; the fact that *L. prolificans* infections are almost exclusively suffered by immunocompromised individuals, although its environmental distribution is associated with humanized areas, seems to indicate that an immunocompetent immune system is able to control the fungus [108,109,110]. Some of the antibodies produced by these individuals may therefore be protective against a possible infection by *L. prolificans*, and the fungal antigens recognized could be therapeutic targets.

#### *Scedosporium*/*Lomentospora* Antigens

Some *Scedosporium*/*Lomentospora* antigens have been identified, including PRM and catalase, which could be interesting as diagnostic markers of these mycoses [51,111]. Among them, catalase A1 from *S. boydii* seems to be a candidate of special interest for serodiagnosis of infections caused by *Scedosporium* spp. in patients with CF [111].

Furthermore, during recent years several immunoproteomics-based studies have been published, enabling identification of a large number of *Scedosporium*/*Lomentospora* targets with potential medical applicability for the production of innovative treatment strategies, diagnosis or prevention of these fungal infections (Figure 4, with references in Appendix A). Along these lines, Buldain et al. (2016) identified the cyclophilin and enolase as the most prevalent antigens of *L. prolificans* recognized by 85 and 80% of saliva from immunocompetent individuals, respectively [109]. These enzymes were also identified on fungal cell wall surface, on the immunomes of *S. apiospermum* and *S. aurantiacum*, and even exhibited cross-reactivity with *A. fumigatus*. These results show that the immune response might offer pan-fungal recognition of these conserved antigens, making them interesting candidates for therapeutic targets, since therapies directed against these enzymes could provide protection against different pathogenic fungi.

In the case of immunocompetent individuals, Pellon et al. (2016) identified the proteins WD40 repeat 2, malate dehydrogenase (Mdh), and DHN1 in conidia of *L. prolificans*, and heat shock protein (Hsp) 70, Hsp90, ATP synthase β subunit, and glyceraldehyde-3-phosphate dehydrogenase (Gapdh) in hyphae, as the most sero-prevalent antigens, detected by at least 90% of the human sera used [108]. Among them, Hsp70, Mdh and WD40 repeat 2 protein were identified in both morphologies. It is worth noting that ATP synthase α and β subunits, fructose-bisphosphate aldolase (Fba), Mdh, enolase and Hsp70 have also been identified as IgA-recognized antigens from saliva of healthy individuals [107,109]. Furthermore, relevant antigens such as Hsp70, enolase, and Hsp90 are present in the surface of fungal cell wall, increasing the interest of these sero-prevalent antigens as therapeutic and diagnostic targets due to their accessibility to the immune system [108].

More recently, Buldain et al. (2019) have demonstrated that sera from mice infected intravenously with *L. prolificans*, *S. apiospermum* and *S. aurantiacum* species, but not with *A. fumigatus*, showed a high cross-reactivity, which is most likely due to the phylogenetic proximity of *Scedosporium*/*Lomentospora* species [112]. This high cross-reactivity makes it impossible to find species-specific antigens inside *Scedosporium*/*Lomentospora*, but it may be very useful for differentiating this complex from *Aspergillus*. Among the antigens recognized by the sera of mice infected with these three species were: in *L. prolificans,* total protein extract, the proteasomal ubiquitin receptor, carboxypeptidase Y (CPY) and vacuolar protein sorting-associated protein 28 (Vps28); in the secretome, a protein of unknown function (Hp jhhlp_006787), a glycosyl hydrolase belonging to the GH16 family (GH16) and a cerato-platanin; and in both extracts, a halo-acid dehalogenase-like hydrolase (HAD-like hydrolase). Secreted proteins are of particular interest because they may be present in higher concentrations in body fluids and they share very little to no homology with human proteins [113,114]. In addition to diagnosis, these antigens may be of especial interest as therapeutic targets for studying alternative therapies for these infections, due to their null or low homology with their human homologues. Among all antigens identified in these studies (Figure 4), Buldain et al. (2019) stressed the importance of *Scedosporium*/*Lomentospora* Hsp70 for its reactivity, prevalence, location, and capacity to differentiate the infections generated by these species from *A. fumigatus*.

In the studies of humoral response carried out so far, a large number of targets of *Scedosporium*/*Lomentospora* with potential medical applicability have been identified, among which Hsp70, carbohydrate portion of PRM and catalase are of particular interest [50,51,107,108,109,111,112,115]. Nonetheless, so far only catalase A1 from *S. boydii* has been tested for clinical usefulness [111]. Although there is as yet no commercially available serological test for *Scedosporium*/*Lomentospora*, interest in the development of such detection systems is increasing. This is reflected in a recently-published article developing a useful ELISA test to detect *Scedosporium*/*Lomentospora*-infected patients and discriminate them from those infected with *Aspergillus* [28].

## 5. Fungal Mechanisms of Immune Response Evasion

The mechanisms of the pathogenic fungi *Scedosporium*/*Lomentospora* for immune response evasion are largely unknown. However, in the studies performed so far, different fungal cell wall components involved in this process have been detected. Among them, the surface glycoprotein PRM of *L. prolificans* has an immunomodulatory effect by reducing the inflammatory response and increasing host non-protective response. Precisely, immunization with PRM in a murine model of invasive scedosporiosis generated a decrease in the secretion of proinflammatory cytokines and chemokines, and an increase in Treg cells and in IgG1 secretion, an immunoglobulin linked to a non-protective response. In this sense, PRM exacerbated the infection process, thereby facilitating colonization, virulence and dissemination of the fungus [103].

Moreover, species of *Scedosporium*/*Lomentospora* produce 1,8-dihydroxynaphthalene melanin (DHN-melanin), also found in *A. fumigatus* conidia [116]. This molecule is known to facilitate evasion of immune response by masking PAMPs, by interfering with phagolysosome formation and acidification and by inhibition of host cell apoptotic pathways [55]. However, *Scedosporium*/*Lomentospora* melanin function has been poorly studied so far. Specifically, Ghamrawi et al. (2014) studied the dynamic changes in *S. boydii* conidia depending on the age of cultures, demonstrating an increase in the amount of melanin over time and suggesting melanin polymerization. This polymerization progressively masks mannose-containing glycoconjugates, which are involved in immune recognition, perhaps allowing the fungus to escape from host immune defenses [116]. In another study, using mutants of *L. prolificans* lacking the dihydroxy-naphthalene (DHN)-melanin biosynthetic enzymes, the authors showed that melanin plays a protective role in the survival of the pathogen to oxidative killing and UV radiation [117]. From the few studies conducted, therefore, it may be concluded that melanin is an important virulence factor of *Scedosporium*/*Lomentospora*, since it masks fungal PAMPs by preventing fungal recognition and decreases oxidative killing by phagocytes [116,117].

On the other hand, it has been detected that *Scedosporium*/*Lomentospora* is capable of destructing host key proteins, which may eliminate host defenses and facilitate tissue invasion. *S. apiospermum* is thus able to efficiently degrade proteins such as the complement factors C3 and C1q in cerebrospinal fluid, presumably by proteolytic degradation [118]. Considering that complement proteins represent a major immune defense of the CNS and that the brain is one of the most affected organs following *Scedosporium*/*Lomentospora* dissemination [9,17], the ability to degrade complement factors might be implicated in the neurotropism of these fungi. An extracellular 33 kDa proteinase of *S. apiospermum* has also been detected which, like alkaline proteinase of *A. fumigatus*, is able to degrade human fibrinogen [119]. Furthermore, mycelia of *S. boydii* releases metallopeptidases capable of cleaving several proteinaceous compounds, including human hemoglobin, IgG extracellular matrix components (fibronectin and laminin) and sialylated proteins (mucin and fetuin) [120].

Enzymes implicated in oxygen species detoxification, catalase and Cu,Zn-superoxide dismutase (SOD), have also been also purified and characterized in *Scedosporium* spp. [121,122]. In particular, it was demonstrated that catalase A1 gene of *S. boydii* is overexpressed in response to oxidative stress and phagocytic cells, whereas the gene encoding SOD (SODC gene) is constitutively expressed [121].

## 6. Conclusions

This review has sought to simplify and explain clearly our knowledge to date on the host immune response against *Scedosporium*/*Lomentospora* species. This response appears to differ in many aspects from that of other fungi and is dependent on the host immune cells involved in the response to each type of infection. The many aspects that remain to be clarified show that more research is essential to shed light on all the immune mechanisms involved in the defense of the host against *Scedosporium*/*Lomentospora*. Among the unknowns that remain to be elucidated, the most important are the complexity of the mechanisms of these species to resist immune response and antifungal treatments, their neurotropism, and their tendency to colonize CF patients. However, studies of these species are very limited and global collaboration between research groups will be crucial to take steps towards understanding these undervalued emerging pathogens.

## Figures and Tables

**Figure 1 jof-07-00075-f001:**
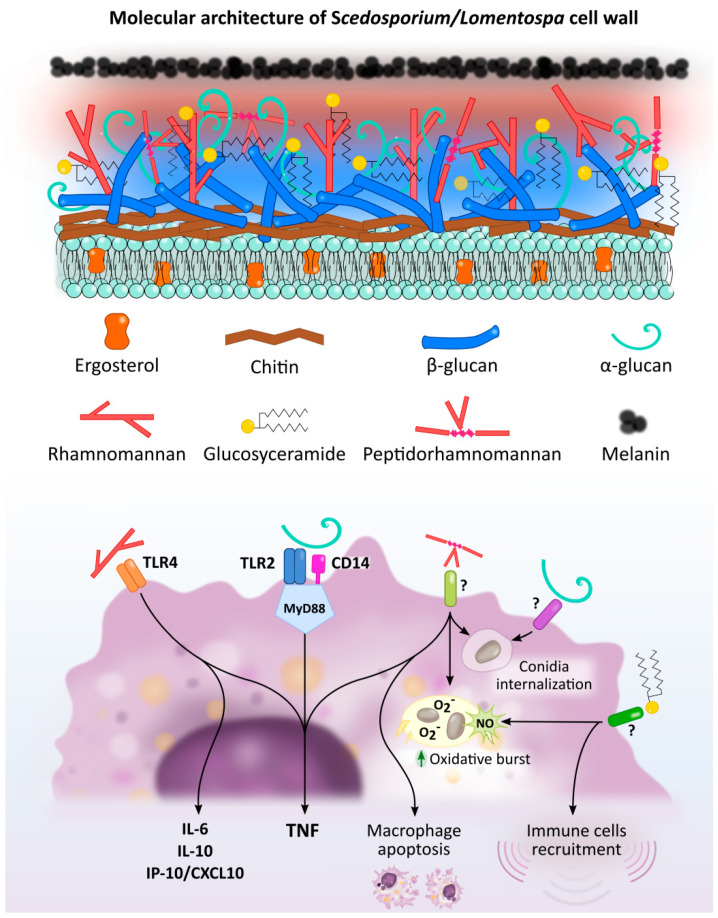
*Scedosporium*/*Lomentospora* cell wall components induced macrophage activation and Pattern Recognition Receptors (PRR) involved in these interactions. Rhamnomannans binds to TLR4 stimulating TNF, IL-6, IL-10 and IP-10/CXCL10 release. α-glucans stimulates TNF release through TLR2, CD14 and MyD88. It is also involved in conidia uptake, but the receptor involved in this process is unknown. Peptidorhamnomannan (PRM) is involved in fungal phagocytosis, induces macrophages death, and stimulates production of TNF, nitric oxide (NO) and superoxide (O_2_^−^). Glucosylceramide (GlcCer) induces NO and superoxide O_2_^−^ production, increases fungal conidia killing, and promotes the recruitment of immune cells, including macrophages. The GlcCer and PRM receptor/s implicated in this activation have not been elucidated.

**Figure 2 jof-07-00075-f002:**
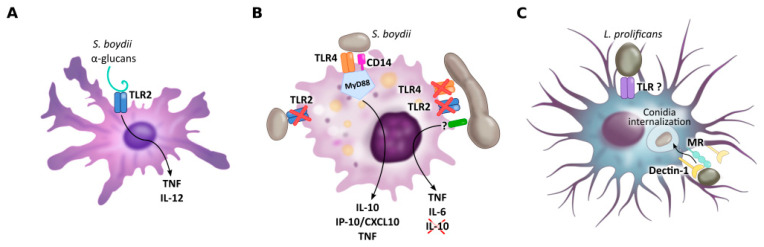
PRRs of dendritic cells (**A**), macrophages (**B**) and microglial cells (**C**) implicated in *Scedosporium*/*Lomentospora* recognition. Dendritic cell TLR2 is necessary for α-glucans recognition and consequent IL-12 and TNF secretion. Macrophage TLR4, together with CD14 and MyD88, but not TLR2, is required to induce macrophage activation by conidia, whereas hyphae recognition is independent of TLR2 and TLR4. Microglial cell mannose receptors, but especially Dectin-1, mediate *L. prolificans* conidia uptake. Other receptors may be involved in this process.

**Figure 3 jof-07-00075-f003:**
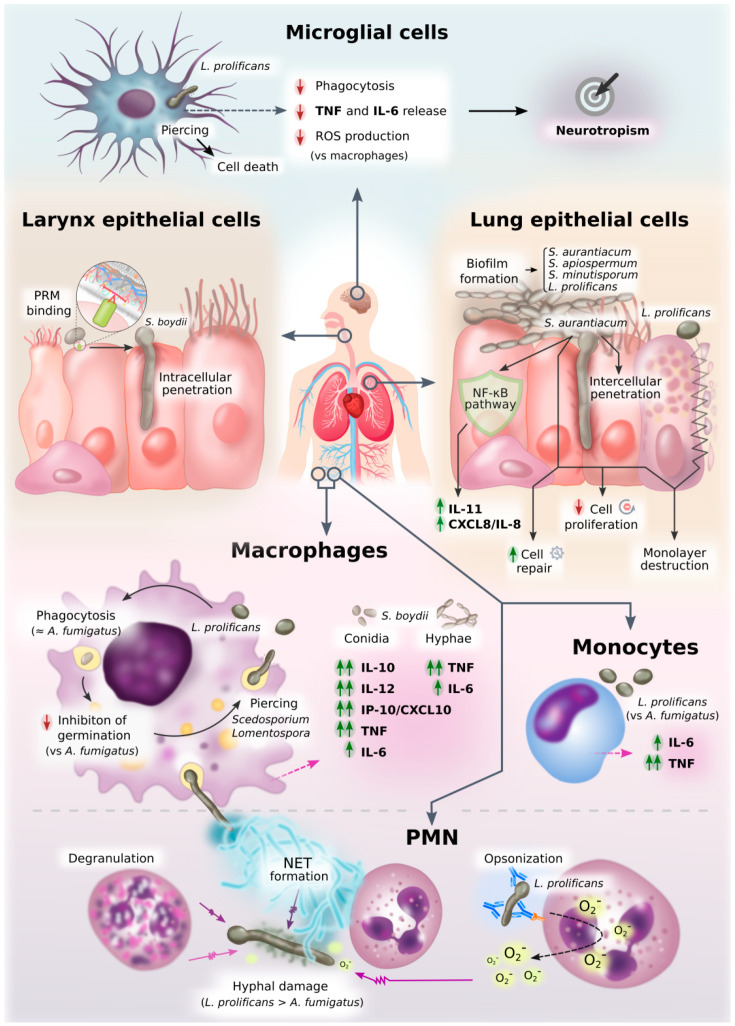
Diagram of innate immune response against *Scedosporium*/*Lomentospora*. This figure graphically represents an overview of all the information on innate cell interactions with *Scedosporium*/*Lomentospora* gathered in this review.

**Figure 4 jof-07-00075-f004:**
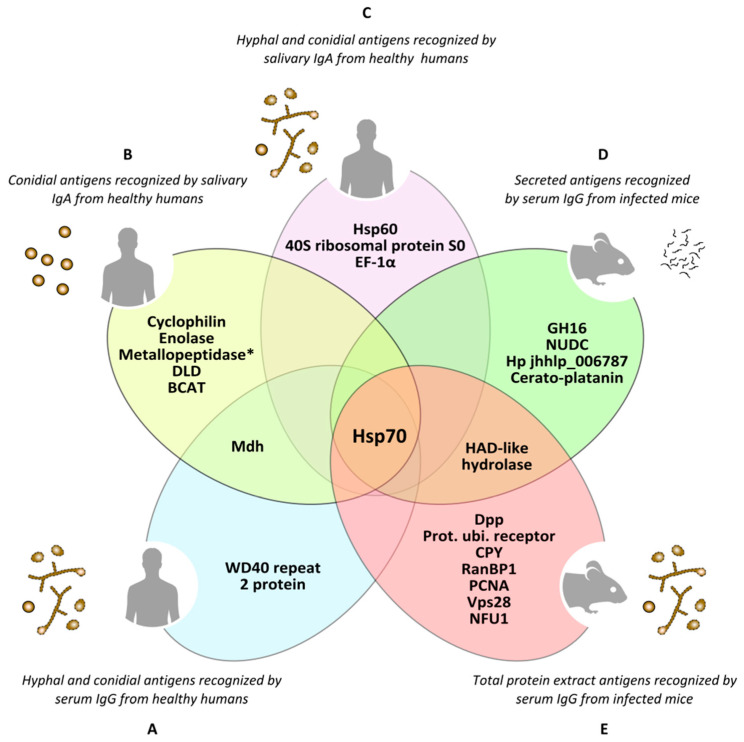
Antigens of *Lomentospora prolificans* identified in several immunoproteomics-based studies. Antigens detected as being reactive against at least 70% of human serum samples in both hyphal and conidia morphologies (**A**). Conidial antigens that reacted with at least 50% of human salivary samples (**B**). The most immunodominant antigens that react with human salivary samples, detected in both fungal morphologies (**C**). The most immunodominant antigens of secretome (**D**) and total protein extract (**E**) recognized by sera from infected mice. Mdh: malate dehydrogenase; Hsp70: heat shock protein 70; Metallopeptidase: Cys-Gly metallo-dipeptidase DUG1; DLD: dihydro-lipoyl dehydrogenase; BCAT: putative branched-chain-amino-acid aminotransferase TOXF; Hsp60: heat shock protein 60; EF-1α: translation elongation factor-1 α; GH16: glycosyl hydrolase family 16 protein; NUDC: nuclear movement protein nudC; Hp jhhlp_006787: protein with unknown function Hp jhhlp_006787; HAD-like hydrolase: halo-acid dehalogenase-like hydrolase; Dpp: dipeptidyl-peptidase; Prot. ubi. receptor: Proteasomal ubiquitin receptor; CPY: carboxypeptidase Y; RanBP1: RanBP1 domain-containing protein; PCNA: proliferating cell nuclear antigen; Vps28: vacuolar protein sorting-associated protein 28; NFU1: NFU1 iron-sulfur cluster scaffold-like protein. * This antigen has been added by repeating the peptides spectra analysis with a later version of the NCBI (National Center for Biotechnology Information) database.

## Data Availability

Not applicable.

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
