# Peer review of "The Host Immune Response to Scedosporium/Lomentospora"

_jof, 2021, doi:10.3390/jof7020075_

Round 1

Reviewer 1 Report

Dear Authors,

The manuscript ID: jof-1066841 entitled “The host immune response to Scedosporium/Lomentospora” written by Idoia Buldain, Leire Martin-Souto, Aitziber Antoran, Maialen Areitio, Leire Aparicio-Fernandez, Aitor Rementeria, Fernando L. Hernando, and Andoni Ramirez-Garcia is devoted to infections caused by the opportunistic fungal pathogens.

The purpose of this review is very interesting, because during the past few decades, opportunistic fungal species of Scedosporium and Lomentospora have become increasingly recognized as a cause of infection, especially in severely ill or immunocompromised and otherwise debilitated patients. The use of mould-active antifungal prophylaxis in these patients has decreased the incidence of invasive fungal disease, but shifted the balance of mould disease to those from non-fumigatus Aspergillus species, Mucorales, and Scedosporium/Lomentospora spp. These infections are recognized to be difficult to treat due to the tendency of these species to exhibit resistance to many commonly used antifungal agents. Moreover, current knowledge on these infections is rather limited. Therefore, understanding the immune response against these pathogens is very important.

According to me, Authors prepared a good manuscript. The knowledge of the host immune response against Scedosporium/Lomentospora species to date has been extensively documented, presented in the form of figures and properly explained. The whole review is appropriately organized and described.

I have only a small suggestion in order to improve paper:

Lines 42, 43, 45, 111, 252, 253, 353: Scedosporium/Lomentospora – please change to italics

Lines 132, 176, 261, 272, 273, 524, 567: L. prolificans – please change to italics

Line 366: in vitro – please change to italics

I think, this review is valuable and may be accepted for the publication in “Journal of Fungi”.

 With highest regards,

Author Response

The purpose of this review is very interesting, because during the past few decades, opportunistic fungal species of Scedosporium and Lomentospora have become increasingly recognized as a cause of infection, especially in severely ill or immunocompromised and otherwise debilitated patients. The use of mould-active antifungal prophylaxis in these patients has decreased the incidence of invasive fungal disease, but shifted the balance of mould disease to those from non-fumigatus Aspergillus species, Mucorales, and Scedosporium/Lomentospora spp. These infections are recognized to be difficult to treat due to the tendency of these species to exhibit resistance to many commonly used antifungal agents. Moreover, current knowledge on these infections is rather limited. Therefore, understanding the immune response against these pathogens is very important.

According to me, authors prepared a good manuscript. The knowledge of the host immune response against Scedosporium/Lomentospora species to date has been extensively documented, presented in the form of figures and properly explained. The whole review is appropriately organized and described.

I have only a small suggestion in order to improve paper:

  • Lines 42, 43, 45, 111, 252, 253, 353: Scedosporium/Lomentospora – please change to italics
    • Corrected
  • Lines 132, 176, 261, 272, 273, 524, 567: L. prolificans – please change to italics
    • Corrected
  • Line 366: in vitro – please change to italics
    • Corrected

 Reviewer 2 Report

Dear authors,

Thank you very much for this interesting review. I have some minor remarks below.

 Abstract

Line 11-12: Should read „..are on the rise.”

Line 15: “them” in the meaning of “these fungi”?

  1. Introduction

33-35: Here it could be nice to explain why fungal infections are increasing (more immunosuppressed patients, more potent immune suppressive drugs) and why fungal resistance is increasing (e.g. resistance of A. fumigatus to azoles due to the use of azoles in agriculture for the protection of crops).

Line 41: Should read “effectivity”.

Line 42-43: Please write Lomentospora and Scedosporium in italic here and throughout the whole manuscript.

Line 64 “suffering from these mycoses”

Line 67 “ranking second among filamentous fungi, after Aspergillus fumigatus”?

Line 73-74 “Treatment is difficult, as these fungi show very limited susceptibility to current antifungals.”?

Line 76-78 Here it has to be noted that combination therapy is only supported by some reports and is therefore only moderately recommended! 

  1. Pathogen recognition

Line 176: Please write L. prolificans in italic here and throughout the whole manuscript.

  1. Activation of innate immune cells

Line 236 The figure legend of Figure 3 is missing

Line 367 “Rhizopus oryzae” has been renamed to Rhizopus arrhizus.

Line 395-397 This is a strange sentence.. I mean one could think it might be the AMB which is effective. Why not “G-CSF treatment combined with liposomal amphotericin improved survival of immunosuppressed mice compared to l-AMB alone, but was not statistically significant”?

  1. Activation of innate immune cells

Line 416: Should read “Activation of adaptive immune cells”

Line 428 Please change first “that” to “than”

Author Response

Abstract

  • Line 11-12: Should read „..are on the rise.”
    • Corrected
  • Line 15: “them” in the meaning of “these fungi”? yes
    • Corrected

Introduction

  • 33-35: Here it could be nice to explain why fungal infections are increasing (more immunosuppressed patients, more potent immune suppressive drugs) and why fungal resistance is increasing (e.g. resistance of  fumigatusto azoles due to the use of azoles in agriculture for the protection of crops).
    • We agree with the reviewer and, therefore, we have included the following paragraph:

“In recent decades, however, the outlook has been changing radically, as fungal infections increase in frequency as a result of a substantial increase in immunosuppressive infections, such as those generated by the human immunodeficiency virus (HIV), as well as in the use of more potent immunosuppressive drugs and in invasive medical interventions. Moreover, resistance to drugs has become more and more common, which may be due to several reasons, including the use of environmental fungicides as reported in Aspergillus resistance to azole.”

  • Line 41: Should read “effectivity”.
    • Corrected
  • Line 42-43: Please write Lomentosporaand Scedosporium in italic here and throughout the whole manuscript.
    • Corrected
  • Line 64 “suffering from these mycoses”
    • Corrected
  • Line 67 “ranking second among filamentous fungi, after Aspergillus fumigatus”?
    • Corrected
  • Line 73-74 “Treatment is difficult, as these fungi show very limited susceptibility to current antifungals.”?
    • Corrected
  • Line 76-78 Here it has to be noted that combination therapy is only supported by some reports and is therefore only moderately recommended!
    • We have included the following sentence:

“However, combination therapy is only supported by some reports and is therefore only moderately recommended.”

Pathogen recognition

  • Line 176: Please write prolificansin italic here and throughout the whole manuscript.
    • Corrected

Activation of innate immune cells

  • Line 236 The figure legend of Figure 3 is missing
    • We have included it:

“Figure 3. Diagram of innate immune response against Scedosporium/Lomentospora. This figure graphically represents an overview of all the information on innate cell interactions with Scedosporium / Lomentospora gathered in this review. ”

  • Line 367 “Rhizopus oryzae” has been renamed to Rhizopus arrhizus.
    • Corrected
  • Line 395-397 This is a strange sentence.. I mean one could think it might be the AMB which is effective. Why not “G-CSF treatment combined with liposomal amphotericin improved survival of immunosuppressed mice compared to l-AMB alone, but was not statistically significant”?
    • Corrected

Activation of innate immune cells

  • Line 416: Should read “Activation of adaptive immune cells”
    • Corrected
  • Line 428 Please change first “that” to “than”
    • Corrected